

# Multi-grained alignment method based on stable topics in cross-social networks

Jing Lu and Qikai Gai

School of Optical-Electrical and Computer Engineering, University of Shanghai for Science and Technology, Shanghai, China

## ABSTRACT

The user alignment of cross-social networks is divided into user and group alignments, respectively. Obtaining users' full features is difficult due to social network privacy protection policies in user alignment mode. In contrast, the alignment accuracy is low due to the large number of edge users in the group alignment mode. To resolve this issue, First, stable topics are obtained from user-generated content (UGC) based on embedded topic jitter time, and the weight of user edges is updated by using vector distances. An improved Louvain algorithm, called Stable Topic-Louvain (ST-L), is designed to accomplish multi-level community detection without predetermined tags. It aims to obtain fuzzy topic features of the community and finalize the community alignment across social networks. Furthermore, iterative alignment is executed from coarse-grained communities to fine-grained sub-communities until user-level alignment occurs. The process can be terminated at any layer to achieve multi-granularity alignment, which resolves the low accuracy issue of edge user alignment at a single granularity and improves the accuracy of user alignment. The effectiveness of the proposed method is shown by implementing real datasets.

## INTRODUCTION

Social networks play an essential role in daily life, such as music sharing, assessment of game services, evaluation of dynamic sharing, *etc.* (*Jiang & Jiang, 2014*). Identifying the same users on different social networks can improve the social experience, increase social efficiency, and make advertising and business recommendations more accurate and understandable. On the other hand, users utilizing several services can be provided by aggregating publicly visible user information from multiple online social networks by linking users' accounts accessible on different social networks. So, their behaviors on multiple online social networks are analyzed to detect and find illegal users trying to steal their private information (*Shi et al., 2023*). At present, cross-social network alignment can be divided into two modes: user alignment and group alignment, respectively.

User alignment is based on user attributes, behaviors, and topology information. *Zeng et al. (2021)* was matched based on the user's profile (user name, age, profile, educational background), UGC, and location attributes by plugging them into the fusion classifier of machine learning. In *Yuan et al. (2021)*, the researchers used the backpropagation (BP)

Corresponding author
Qikai Gai, gaiqikai@163.com

neural network to calculate the similarity of the user name by vector mapping. The alignment method based on users' attributes faces the problems of limited data acquisition and high data processing difficulty. A joint framework of behavior analysis and social network alignment was proposed in *Ren et al. (2020)*. The principle was to obtain comprehensive user behavior information, through a user's behavior such as forwarding, commenting, and location punching information based on a fusion algorithm using earth-moving distance (EMD), thus, user behavior prediction is verified, and user alignment based on behavior characteristics is completed (*Jiang et al., 2022*; *Shi et al., 2018*; *Nie et al., 2016*; *Chen et al., 2020*). However, user behavior information is complex and highly sparse, making it difficult to get potential consistent patterns out of it. In *Reuter et al. (2015)*, they employed anchor users to walk outward to obtain local topological maps around anchor users. It performed a common maximum subgraph similarity calculation or probability-based graph calculation to complete alignment (*Wang et al., 2019*). However, this kind of matching method for pure topology structure relies too much on the influence of anchor users, and the location of anchor points in the network is particularly important. If it is too dense, the local matching calculation will converge quickly, fall into local optimal, and cannot be spread to distant locations. If it is too sparse, the connection between local areas will be weakened, the matching calculation amount will be larger, and the accuracy will be reduced.

On the other hand, group alignment is usually combined with user interest. Researchers in *Jiang et al. (2022)* analyzed the topics of users' published content through the Dirichlet model. It combined the method of hypertext topic search (HITS) (*Shi et al., 2018*). The researchers in *Nie et al. (2016)* proposed that the interest should be divided into temporary and core interests, reduce the interference of temporary interests, and effectively improve alignment accuracy. *Chen et al. (2020)* proposed that the connection between time and UGC should be strengthened, and the alignment should be carried out for the user's single-layer interest community. The authors in *Zheng et al. (2019)* utilized users' interest labels for weight scoring and stratified community classification. Still, the label classification was fixed, and the same user was assigned to multiple label classes, resulting in redundancy. The deep learning model, called ReinCom, was successfully used in *Ding et al. (2021)* for the first time to map nodes and communities to the hyperbolic space to learn the degree of coupling between nodes and communities. However, because the input and output parameters of a hierarchical community tree were difficult to define as integer numbers with one-hot vectors, the updating and optimizing parameters of a community tree varied, and the effect was not good when facing larger communities.

To sum up, most scholars focused on one of the two matching modes of groups or users, and data acquisition was often limited in the user alignment mode. In contrast, in the group alignment mode with determined granularity, there are too many edge users and only reflect the overall characteristics of a group, and accurate user alignment cannot be completed by using this result. At the same time, the single-layer community detection cannot accurately reflect the user relationship interests, and the fixed-label hierarchical detection will cause the same user to assign multiple communities, resulting in redundancy and low efficiency.

The contributions of the article are expressed as follows:

(a) A stable topic acquisition method is proposed. The original keywords of the published content by users are mined, and the original theme sequence is formed by clustering. The jitter degree of each theme is calculated according to the time slice and sorted out. The short-time theme is eliminated and the stable theme sequence is obtained. The user's theme sequence is embedded into the Word2Vec model, and the user's theme similarity is calculated.

(b) A multi-level community detection method based on ST-L is proposed. The topic similarity is used to update the user's edge weights, and the Louvain algorithm is improved based on the weights to optimize the aggregation and re-optimize the upper layer. The multi-level community detection without presetting fixed hierarchical labels is completed.

(c) An improved fuzzy center clustering method is proposed to obtain the community theme features. By updating and optimizing the correlation degree matrix between users and theme feature points, the theme features of communities and user groups are obtained. The theme feature similarity across social networks of the same granularity is calculated to complete alignment by starting from coarse-grained communities. Vertical iteration is performed until user matching is performed. The alignment results of each layer can be output to achieve multi-granularity alignment.

(d) It is concluded that multi-grained alignment's (MGA's) multi-granularity alignment method based on the combination of the two alignments on the real data set has better performance indicators.

# BASIC DEFINITIONS

## Symbol definition

Definition 1 (social network). Let $G = (V, E, P)$ and V define the social network. Let the collection of user nodes be aligned denoted by $V = (u_1, u_2, \ldots u_i, \ldots u_R)$, where i denotes the user numbers. E stands for powerless edge set $E = \{e_{ij}|u_i, u_j \in V\}$ and $e_{ij}$ stands for users $u_i$ and edges; that is, there exists a followed relationship between users; that is, there is a connected edge. $u_j$ P stands for the content published by the user denoted by $P = $ (user, published content, published time), indicating the content set published by the user $P_i$ $u_i$. For the convenience of subsequent modeling and calculation, the source social network and target social network are selected and denoted $G = (V, E, P)$ $G' = (V', E', P')$, respectively.

Definition 2 (thematic hierarchical communities). Let a topic-layered community be denoted by $C = \left(c_1^{00}, c_2^{01}, \ldots c_1^{10}, \ldots c_i^{jh}, \ldots c_R^{L}\right)$ where $i$ represents the community to which the user belongs, $u_i$ $j$ represents the layer of the community, $h$ represents the number of the community, and when $j = 0$, the community is called a user group.

Definition 3 (user topics). Let the user topic and the topic weight be denoted by $topic = (topic_1, topic_2, \ldots topic_R)$ $W = w_1, w_2, \ldots w_R$, where the topic weight subscript corresponds to the topic subscript.

Definition 4 (anchor users). An anchor user is someone who has an account on both the source social network and the target social network and has the account aligned, authenticating high-impact users by name denoted as $G$ $G'$.

Definition (five thematic characteristics). The fusion topic feature of the user group and community is defined as representing the overall topic feature of the user group or the members of the community in layer j numbered h, where the feature point of the middle topic z represents the number of the feature points. The dimension of the feature point should be the same as the dimension of the user topic vector denoted as:

$$V_{c^{jh}} \ CTopic_z^{V_{c^{jh}}} \ V_{c^{jh}}.$$

## The problem definition

When the source of the social network is a concern, the target social network is found with a stable theme to detect and find the user group and the high-level community. Then, the sum of the theme features is obtained from the coarse-grained high-level community to the fine-grained community to reach the user-level alignment denoted as:

$$u \in G \ u' \in G' \ u \ u' \ c_u^{jh} \ c_u^{jh'} \ V_{c^{jh}} \ V'_{c^{jh}} \ u \ u'.$$

# STABLE TOPIC MULTI-GRANULARITY ALIGNMENT METHOD: MGA

## The framework of the method

The steps of the multi-grained alignment (MGA) method are shown in Fig. 1. The steps are summarized as follows:

Step 1: For source network and target network users, the time jitter degree of the theme is calculated from the original topic, the unstable hot topics are filtered out, the stable topics are obtained and embedded into the same vector space, and the similarity of the user topic is calculated.

Step 2: The user weights are updated according to the user theme similarity, the Louvain algorithm is improved without weights, community detection is conducted and iterated from low to high levels, and a hierarchical community structure is formed without preset fixed labels.

Step 3: The fuzzy centre clustering algorithm is improved to extract the theme features of user groups or communities by optimizing the correlation degree matrix and theme feature points. According to the similarity of theme features, cross-social network alignment is carried out from coarse-grained to fine-grained by running user-level alignment iteratively, and the alignment results of each layer can be used as output.

## The extraction of a stable topic

### The original theme

For the blog sets published by users, the Term Frequency-Inverse Document Frequency (TF-IDF) model is adopted to extract the sequence of users' interested keywords, and K-means clustering (*Macqueen, 1967*) is carried out. Centroid words of each cluster are taken as the subject words, K original topics and their weights are obtained, and the users' original interest matrix is formed, as shown in Eq. (1).

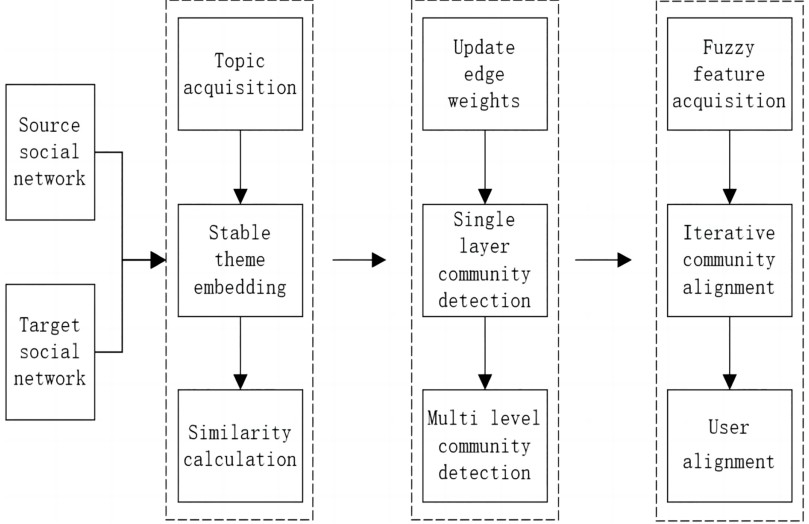

**Figure 1 Flow chart of MGA.**

$$A_{u_i} = \begin{pmatrix} w_1 \cdot topic_1^{u_i} \\ w_2 \cdot topic_2^{u_i} \\ \vdots \\ w_k \cdot topic_k^{u_i} \end{pmatrix}.$$

(1)

### The filter-out of short-time hot topics

Users are extremely vulnerable to the impact of hot events, and their attention to them is relatively concentrated (*Nie et al., 2016*) in time, which does not mean that users can stabilize the topic. Therefore, this article will find and filter out short-time hot topics with large jitter degrees according to the time slice.

First of all, the weights of the medium are decomposed based on the total number of time slices $A_{u_i}$ $topic_j$ $w_j$ $T$ and the length of each slice t, and the weights of the user topic in each time slice are obtained. The time jitter $St$ of the user topic is defined in Eq. (2).

$$St_{topic_j}^{u_i} = \frac{1}{T} \times \sum_{t=1}^{T} \left( w_{j,t}^{u_i} - \frac{1}{T} \times \sum_{t=1}^{T} w_{j,t}^{u_i} \right)^2$$

(2)

where $w_{j,t}^{u_i}$ represents the weight of the user's topic in the $u_i$ $t$ time slice, $T$ represents the total number of time slices, and $t$ represents the number of time slices.

According to Eq. (2), the time jitter of all the *users' topics* is obtained, which is arranged in ascending order. The higher the degree of the topic, the lower the weight is, and it is called the short-term hot interest that needs to be removed. The top $kt$ topics are selected and the topics with a large jitter degree are cut. For statistical purposes, the weight of the reduced unstable topics is redistributed proportionally. Finally, the users' topic matrix is obtained as shown in Eq. (3), and the filtering process is presented in Algorithm 1.

**Algorithm 1 The filter-out the user's short-time topic.**

Enter: $A_{u_i}$, $w_{j,t}^{u_i}$

Output: $A_{u_i}{'}$

a) St(j)← $St_{topic_j}^{u_i}$, topic[]← $A_{u_i}$

b) For topic[j] in topic[] do

c) For t in T do

d) St(j)←St(j)+()// Calculate jitter $w_{j,t}^{u_i} - avg(w_{j,t}^{u_i})$

e) St(j)←St(j)/T

f) Order by St asc

g) For j in (kt,k] do

h) Then for q in [0, kt] do

i) $w_q$ ←*(1+) // Assign cut edge weight $w_q$ $w_q * w_j$

j) Update $w_{q'}$ ← $w_q$

k) Return $A_{u_i}{'}$

$$A_{u_i}{'} = \begin{pmatrix} w_1{'} \cdot topic_1^{u_i} \\ w_2{'} \cdot topic_2^{u_i} \\ \vdots \\ w_k{'} \cdot topic_{kt}^{u_i} \end{pmatrix}. \tag{3}$$

### The calculation of users' topic similarity

The user, $u_i$, topic matrix is extracted and embedded into the $A_{u_i}{'}$ Word2Vec (*Mikolov et al., 2013*) model based on Chinese Wikipedia training, and weighted summation is employed to generate the user topic vector, as shown in Eq. (4).

$$V_{u_i} = w_1{'} \cdot \overrightarrow{topic_1^{u_i}} + w_2{'} \cdot \overrightarrow{topic_2^{u_i}} + \cdots w_{k'}{'} \cdot \overrightarrow{topic_{k'}^{u_i}} \tag{4}$$

According to the vector sum of users that utilize the theme and $u_i$ is obtained from Eq. (4), the similarity of the theme vector is calculated to obtain the user's theme similarity, as shown in Eq. (5): $u_i$ $u_j \in U$ $V_{u_i}$ $V_{u_j}$ $Sim\_Topic(u_i, u_j)$

$$Sim\_Topic(u_i, u_j) = \frac{V_{u_i} \cdot V_{u_j}}{|V_{u_i}| \cdot |V_{u_j}|} \tag{5}$$

## The hierarchical detection algorithm of user-stable topic: ST-L

### Data preprocessing

For a given social network $G$, there exists an edge between users in the original network diagram, which is a concerned relationship and cannot reflect the degree of correlation between the two subjects denoted as $u_i$ $u_j$. To better detect the hierarchical community of

user topics, this article converts user edge $E$ into a weighted edge, takes the user topic similarity used in Eq. (5) as the edge weight between users, and then transforms social network $\Delta E \; Sim\_Topic(u_i, u_j) \; u_i u_j \; G$ into a weighted undirected network graph of users' topics.

### The detection method of users' hierarchical community topic

The Louvain (*Blondel et al., 2008*) algorithm is an excellent hierarchical community detection method, that has leading performance in large-scale complex networks, but users without rights affect its effect in special application scenarios. To better detect topics in the hierarchical community, this article updates the user's edge weights according to the similarity of the users' topics. The Louvain algorithm (ST-L) is proposed based on stable topics and is divided into two parts: community optimization and community aggregation, respectively. The algorithm assesses which community is more suitable for the nodes in the network based on the community modularity. $Q$, community modularity, is shown in Eq. (6):

$$Q = \frac{1}{2m} \cdot \sum_{u_i, u_j \in U} \left[ e'_{ij} - \frac{d_i \cdot d_j}{2m} \right] \delta(c_i, c_j) \tag{6}$$

where Eq. (6) represents the topic similarity between users, the sum of the theme similarity of all users when connected, respectively, and belongs to, $c_i = c_j \; \delta(c_i, c_j) = 1 \; \delta(c_i, c_j) = 0$ $e'_{ij} \; u_i \; u_j \; d_i \; d_j \; u_i \; u_j \; c_i, c_j \; u_i \; u_j$, $m$ represents the sum of the similarity between all users in the network. The larger the value is $[-1, 1]$, the better the structure is:

The core of the algorithm is to compare the modularity of a user before and after joining a community, and the modularity increment is derived from Eq. (6) and presented in Eq. (7): $u_i c_i \; \Delta Q$

$$\begin{aligned}
\Delta Q &= \left[ \frac{d_{in} + d_{i,in}}{2m} - \left( \frac{d_{ci} + d_i}{2m} \right)^2 \right] - \left[ \frac{d_{in}}{2m} - \left( \frac{d_{ci}}{2m} \right)^2 - \left( \frac{d_i}{2m} \right)^2 \right] \\
&= \frac{1}{2m} \left( d_{i,in} - \frac{d_{in} \cdot d_{i,in}}{m} \right)
\end{aligned} \tag{7}$$

where Eq. (7) presents the sum of the similarity of users in the community, the sum of the similarity of users connected to other users in the community, and the sum of the similarity of all users connected to each $d_{in} \; c_i \; d_{i,in} \; u_i \; c_i \; d_{ci} \; c_i$, respectively.

After obtaining the modularity increment, community optimization can be carried out. A different user group is initialized for each user on the network $\Delta Q$. When partitions are initialized, the number of user groups will be as large as the number of users. Then, for each user and its neighbor users, the modularity increment is calculated by placing it in the user group that will be removed from its user group and placed in the user group with the largest modularity increment $u_i \; u_j \; u_i \; u_j \; u_i$. If so, the original user group stays there. When $\Delta Q < 0$ this process is repeated and applied sequentially for all users (the ordering of users has no significant effect (*Blondel et al., 2008*) on the obtained modularity increments) until completed.

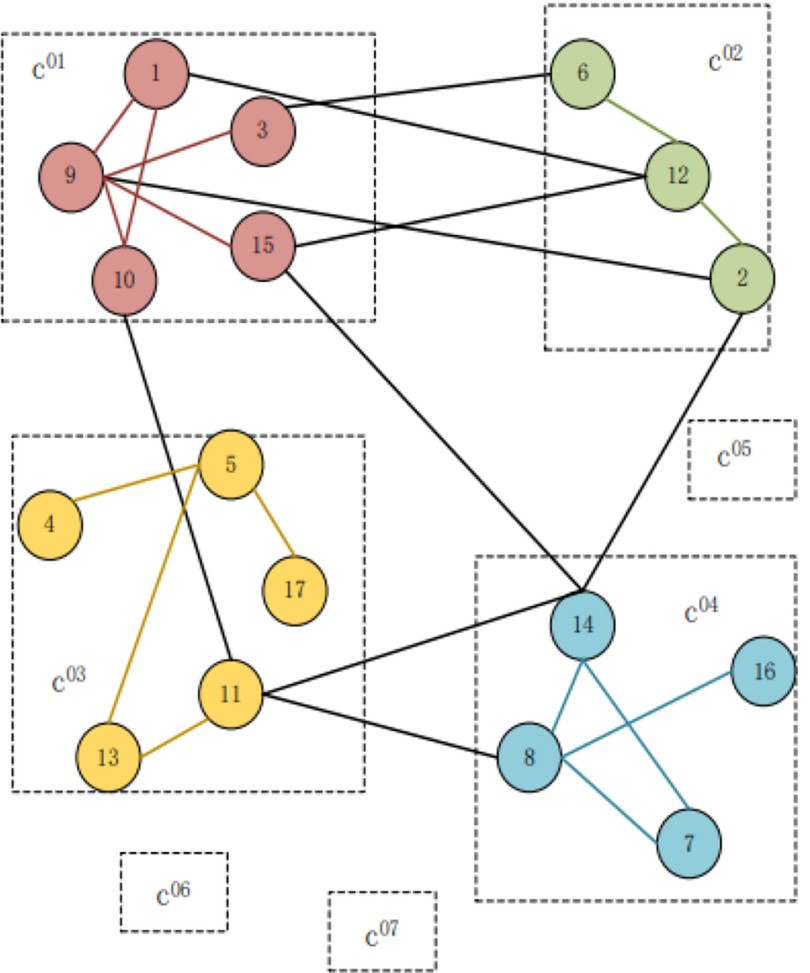

**Figure 2 Examples of single-layer community detection.**

Figure 2 depicts that seven user groups, $c^{0x}$, are supposedly obtained from the first part of the algorithm where x denotes the group serial number such as $c^{01}$: {1,3,9,10,15}, $c^{02}$: {2,6,12}, $c^{03}$: {4,5,11,13,17}, $c^{04}$: {7,8,14,16}. Each number represents a user; the edge indicates that there is a relationship between users to be followed. The users in the same dotted box are considered to be divided into the same group, indicating that the user theme is similar. The structure in groups $c^{05}$, $c^{06}$, and $c^{07}$ and the connection relationships with other groups are omitted for the sake of simplification.

The next step is group aggregation; that is, the user group obtained in the first part is iterated upward, and the weight of the edge between them is the sum of the weight of the edge between the nodes in the corresponding two groups. Then, the algorithm for the first part is iterated on this new graph. After the algorithm of the first part is completed, the second part can be employed to generate a higher-level structure graph. The MGA adopts a three-layer structure, and the results of the top three layers are sparse (*Blondel et al., 2008*).
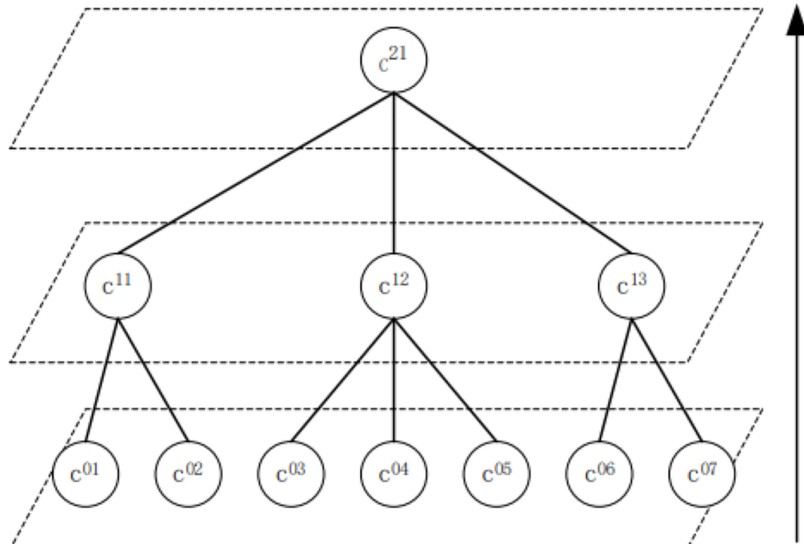

**Figure 3 Examples of multi-layer community detection.**

Figure 3 depicts that after the first part of the algorithm is completed, the second part is iterated to the upper layer. The first iteration is presented as follows: for the user group $c^{0201}$, $c^{03}$, $c^{04}$, $c^{12}$, and $c^{07}$ communities $c^{11}$, $c^{05}$, $c^{06}$, and $C^{13}$, respectively. The second iteration of community $c^{11}$, $c^{12}$, $c^{13}$ constitutes community $c^{21}$.

## The alignment of a multi-granular community with users

### The extraction of topic features

For the communities in the source social and the target social networks, the two communities can be regarded as the anchor communities denoted $G$ $G'$ (*Sun et al., 2020*) after they are aligned. The communities connected with the anchor community pair are respectively in the middle and are called the connected community pair, that is, the community pair to be aligned denoted $G$ $G'$.

After the first round of community detection is completed, users form a group, continue to conduct community detection, and aggregate into the upper community. In this process, each group will produce a group theme feature because it is close to the group member theme vector. The user theme vector is also very close to the community theme feature points, and there are a large number of edge points, which are in the middle of the two feature points. If it is directly forced to be classified into one class and the feature points are updated at this distance, the accuracy will be reduced $V_{cjh}$. The fuzzy centre-based clustering (FCM) algorithm (*Naderipour, Fazel Zarandi & Bastani, 2022*) proposed the idea of fuzzy classification, and the clustering result was expressed as the sample belonging to all kinds of probability matrices, which effectively improves the classification effect of edge users. Therefore, the article introduced the correlation degree matrix between the user theme vector and the community feature points, as shown in Eq. (8).

$$S^\beta = \begin{pmatrix} d\left(V_{u_1}, CTopic_1^{V_1^{jh}}\right), \cdots d\left(V_{u_1}, CTopic_z^{V_1^{jh}}\right) \\ d\left(V_{u_2}, CTopic_1^{V_2^{jh}}\right), \cdots d\left(V_{u_1}, CTopic_z^{V_2^{jh}}\right) \\ \vdots \\ d\left(V_{u_a}, CTopic_1^{V_a^{jh}}\right), \cdots d\left(V_{u_a}, CTopic_z^{V_a^{jh}}\right) \end{pmatrix}. \tag{8}$$

The correlation matrix is optimized and updated by a gradient descent algorithm to obtain the best features $V_{c^{jh}}$. It is a multi-dimensional topic vector, which can represent the theme features of a whole group or community to the greatest extent $V_{c^{jh}}$. Equation (9) presents it.

$$V_{c^{jh}} = \begin{pmatrix} CTopic_1^{c_i^{jh}} \\ CTopic_2^{c_i^{jh}} \\ \vdots \\ CTopic_z^{c_i^{jh}} \end{pmatrix}. \tag{9}$$

To achieve the optimality, the objective function presented in Eq. (10) is constructed as $V_{c^{jh}}$.

$$f(\beta) = \begin{cases} \sum_{a=1}^{N} \sum_{b=1}^{Z} S_{ab}^\beta \left\| V_{u_a} - CTopic_b^{c^{jh}} \right\|^2 \\ \sum_{b=1}^{Z} S_{ab}^\beta \geq 1 \end{cases}. \tag{10}$$

where the user is the weighted constant, the topic vector of the user, and a topic feature point in the topic feature are denoted by $\beta$ $u_a(1 \leq a \leq N) \in c^{jh}$ $V_{ua}$ $u_a$ $CTopic_1^{c_i^{jh}}$ $c^{jh}$ $V_{c^{jh}}$, respectively, $b$ is the number of the feature points, indicating the correlation degree between the user's topic vector and the feature points, the problem is transformed into a minimum problem $S_{ab}^\beta f(\beta)$. The sum of the correlation degree of each user's theme vector is adjusted to the condition that it is greater than or equal to 1, and arranged to attain Eq. (11) as shown $u_a$ $V_{ua}$ $CTopic_1^{c_i^{jh}}$.

$$f(\beta) = \sum_{a=1}^{N} \sum_{b=1}^{Z} S_{ab}^\beta \left\| V_{u_a} - CTopic_b^{c^{jh}} \right\|^2 + \sum_{b=1}^{Z} \mu_b \left( 1 - \sum_{a=1}^{N} S_{ab}^\beta \right) \tag{11}$$

where $\mu_b$ represents any coefficient and the partial derivative of Eq. (10) is equal to 0, and its iterative equation is obtained as shown in Eqs. (12) and (13) $S_{ab}^\beta$ $CTopic_b^{c^{jh}}$.

$$S_{ab}^\beta = \left( \sum_{k=1}^{Z} \left( \frac{\left\| V_{ua} - CTopic_b^{c^{jh}} \right\|}{\left\| V_{ua} - CTopic_k^{c^{jh}} \right\|} \right) \right)^{-1} \tag{12}$$

> **Algorithm 2** The generation of topic features.
>
> Input: $V_{ua}$ (User $u_a(1 \leq a \leq N) \in c^{jh}$)
>
> Output: $V_{c^{jh}}$
>
> a) Initialize V = $[V_{c^{jh}}]$ matrix, V(0)
>
> b) Calculate S = $[S_{ab}^{\beta}]$ matrix, S(0) with $V_{ua}$
>
> c) k-step: calculate the vectors CTopic(k)=[] with S(k) by Formula (13) // Update the topic point $CTopic_b^{c^{jh}}$
>
> d) Update S(k),S(k+1) by Formula (12) // Update the correlation matrix
>
> e) If $\|S(k+1) - s(k)\| < \varepsilon$
>
> f) Then stop
>
> g) Else return to step c
>
> h) Return $V_{c^{jh}}$

$$CTopic_b^{c^{jh}} = \frac{\sum\limits_{a=1}^{N} S_{ab}^{\beta} \cdot V_{ua}}{\sum\limits_{a=1}^{N} S_{ab}^{\beta}} \tag{13}$$

where k denotes the number of iterations. When the iteration is terminated, it means that the iterative change of the user theme vector and the correlation degree of feature points is small, and no further optimization is required. Algorithm 2 presents the steps of $\nabla S_{ab}^{\beta} < \varepsilon$.

### The alignment method of multi-granular community

For the communities and groups in source social network $G$ and target social network, the topic features are obtained in the same vector space based on Algorithm 2, and pair-to-pair alignment is carried out at the same level, as shown in Eq. (14) $G'$.

$$Sim\_V_c = \frac{\sum\limits_{b=1}^{z}\left(CTopic_b^{c^{jh}} \times CTopic_b^{c^{jh}\prime}\right)}{\left|\sum\limits_{b=1}^{z} CTopic_b^{c^{jh}}\right| \times \left|\sum\limits_{b=1}^{z} CTopic_b^{c^{jh}\prime}\right|}. \tag{14}$$

Because the calculation is conducted on the same dimensional matrix, the distance operation with $()V_{c^{jh}} \ V_{c^{jh}}{}' \ V_{c^{jh}} \ V_{c^{jh}}{}'$ T can be performed to obtain a square matrix; the module of the square matrix is the similarity of two communities or two user groups, respectively.

This article adopts the alignment method of multi-granularity iteration from the upper large community with coarser granularity to the lower level concerning running order. Figure 3 shows the community structure. Figure 4 depicts that $C^{21}$ and $C^{21}$ are the highest-level communities and have completed the alignment. When the iteration reaches 21, the insides of $C^{21}$ and $C$ are aligned with their internal sub-communities. In this round, $C^{1212}$ and $C$ are aligned, and the iteration continues to its internal user group. The user group $C^{05'05}$ is aligned with $C$ and continues iterating into two user groups for user matching.

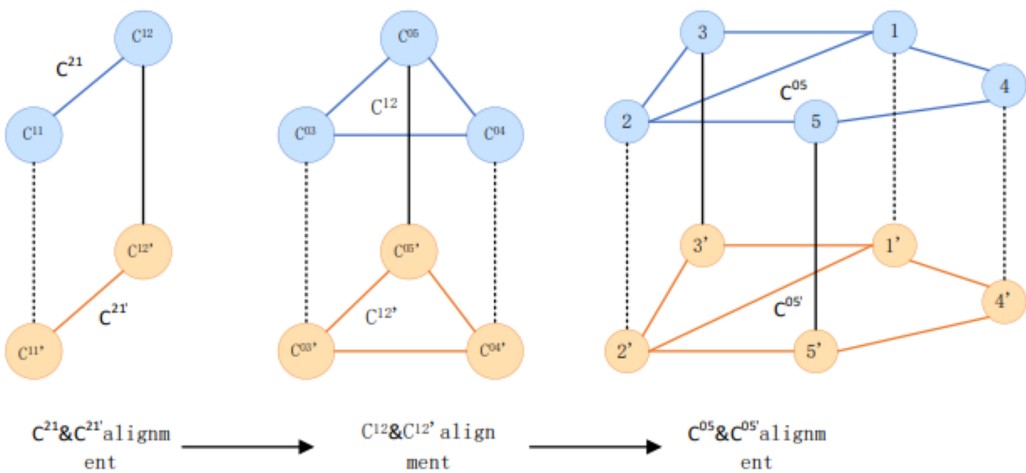

**Figure 4 Examples of multi-grained community alignment.**

### User alignment

When user groups are aligned in the source social network *G* and the target social network, user members can match users based on users' attribute data. For two groups of small users with highly similar themes, the attribute difference is an efficient way to distinguish them *G'*. Similar user names (*Yuan et al., 2021*) and user profiles such as profile, education, unit, *etc.* (*Zeng et al., 2021*) are important representations of users' characteristics. Therefore, based on the document similarity algorithm (*Kusner et al., 2015*), this article employs the combination of user name similarity and personal data similarity to treat users and score matched users. Equation (15) gives the calculation of user attribute similarity.

$$Sim\_A(u, u') = \min_{P \geq 0} \sum_{i,j=1}^{n} P_{ij} D_{ij} \tag{15}$$

where Eq. (15) represents the best conversion matrix and the word shift distance $P_{ij} D_{ij}$.

## EXPERIMENT AND ANALYSIS

### Data sets

The article utilizes a crawler to obtain the user data set from Weibo-Zhihu, which is a real network to run experiments. When the data is acquired, users with less than 20 posts per year are filtered out to avoid the topic being inaccurate due to the sparsity of USG. By identifying Weibo accounts or high-impact real-name authentication anchor users in Zhihu accounts, 1,468 pairs of positive samples were obtained. In addition, 1,468 pairs of non-aligned users were randomly captured in Zhihu and Weibo as negative samples, so that the number of positive and negative samples was equal. In the end, a total of 5,876 users were obtained, including 61,002 users' contacts and 138,749 users' posts. In the experiment, to extract the topic of the user, the "Harbin Institute of Technology stop Word List" (*Guan, Deng & Wang, 2017*) assisted filtering to avoid interference.

To analyze the performance of the subsequent community detection, this article also analyzed the external performance indicators of the community by using the open-labelled data set Aminer (*Ding et al., 2021*) and Cora (*Sun et al., 2019*), respectively. Aminer and Cora contain 12,840 users, 190,658 edges, four preset categories, and 2,708 users, 5,278 edges, and seven default categories, respectively.

## Evaluation metrics

### Evaluation index of the original and stable topic number of users

The contour coefficient is used to evaluate the clustering effect of user topics represented by Eq. (16).

$$s(i) = \frac{b(i) - a(i)}{\max(a(i), b(i))} \tag{16}$$

where Eq. (16) represents the average distance between $a(i)$ $i$ and other samples of the same cluster and the minimum average distance between $b(i)$ $i$ and other clusters, respectively.

Topic jitter time is used to evaluate the effect of stable topic selection, as shown in Eq. (16).

### Evaluation index of feature points

Feature density (*Naderipour, Fazel Zarandi & Bastani, 2022*) is used to evaluate the effect of feature extraction on community themes represented by Eq. (17).

$$\Delta dense = avg\left(\frac{|\{(p,q)|p,q \in in_i, (p,q) \in in_i\}|}{|\{in_i\} + \{out_i\}|}\right) \tag{17}$$

where Eq. (17) represents the sample point, the edge collection of feature points within the link cluster, and the edge collection of feature points outside the link cluster $p, q$ $in_i$ $out_i$, respectively.

### Evaluation index of community detection

Normalized mutual information (NMI) and modularity indexes are mainstream indicators (*Wang, Hao & Guan, 2020*) used to evaluate the performance of community detection, respectively. They are presented in Eqs. (18) and (19).

$$NMI = \frac{2 * I(U, V)}{H(U) + H(V)} \tag{18}$$

where $U$, $V$, $I$, and $H$ represent real classification, the results of the classification, interactive information, and cross-entropy, respectively.

$$modularity = \sum_{i=1}^{n} \left[\frac{L_i}{TL} - \left(\frac{D_i}{TL}\right)^2\right] \tag{19}$$

where $L_i$ represents the total number of edges in the community, $i$ is the sum of vertex degrees in the community i, $D_i$ $TL$ is the total number of edges, $NMI$ measures the

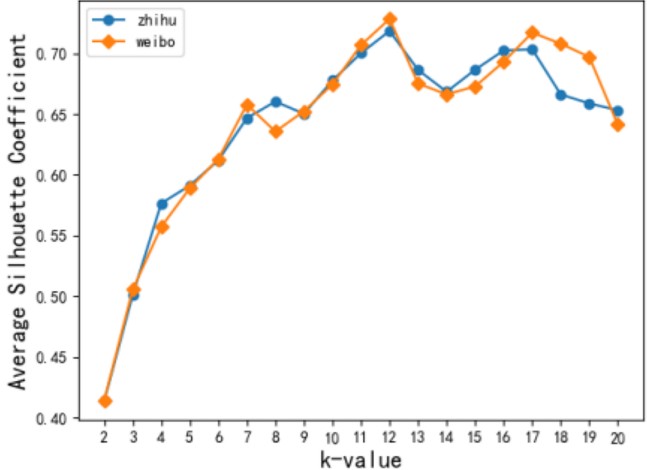

**Figure 5 Effect of k value on contour coefficient.**

similarity between the algorithm's running result and the preset label. Modularity measures the inner tightness of the community.

### The evaluation indicators of the alignment performance

The Accuracy index is used to detect alignment accuracy (*Chen et al., 2017*) across network communities and user groups, represented by Eq. (20).

$$Accuracy = \frac{1}{NC} \sum_0^j \sum_0^h success\left(c^{jh}, c^{jh\prime}\right) \qquad (20)$$

where NC represents the total number of communities and user groups; if the alignment of users exceeds 20% of the total number, one is returned. Otherwise, 0 is returned $c^{jh}, c^{jh\prime}$ $success\left(c^{jh}, c^{jh\prime}\right)$.

The metrics used for user alignment accuracy are called precision, recall, and F1 values, respectively.

## Experimental results and analysis

### Analysis of users' original topic number k and users' stable topic number kt

In this article, the TF-IDF model was used to analyze users' interests, and then 50 keywords with the highest frequency and at least ten occurrences were selected after the stop words were removed. K-means clustering was performed on the keyword sequence to generate *K* original topics. Figure 5 shows the change in the average contour coefficient when the value of *k* topic numbers changed from [2, 20]. Figure 6 depicts that both Zhihu and Weibo users reach the local optimal average contour coefficient when k takes a value of 12. In subsequent experiments, users of the two networks are processed together to analyze parameter selection, and the original topic takes a value of 12 to carry out a short-time hot spot filtering discussion.

When short-term hot topics are filtered out, the change of the statistical trend for users' topics tends to be flat (*Ni, 2020*) according to monthly observations. Therefore, the time slice t in this article takes a month as a unit, and a total time length of 12 months is selected.

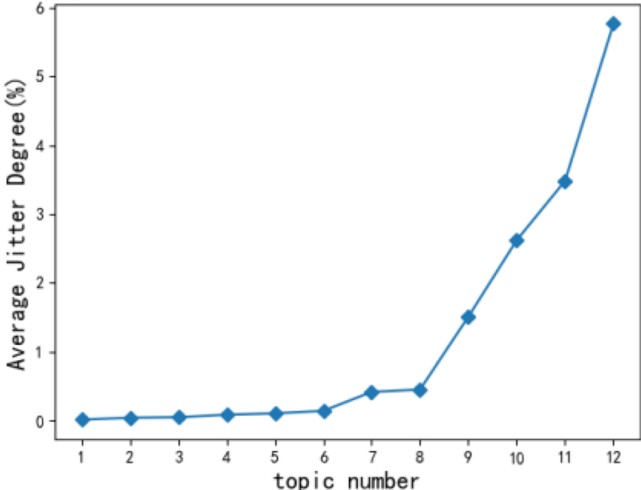

**Figure 6  Average dithering degree of topic.**     

According to Eq. (2), the jitter degree of the 12 original topics of users is calculated and arranged in ascending order. Figure 6 shows the average jitter degree of users' original topics. The average jitter degree of the 9th topic is significantly higher than that of the 8th topic. Concluded that $kt = 8$ can achieve the optimal experimental state, and the user topic becomes the most stable.

### Z-analysis of community theme feature points

The number of feature points $z$ for the theme feature is related to the accuracy of describing the community theme features. If the $z$ value is too small, the feature is too general and not accurate enough; on the other hand, if it is too large, it is too sparse. In this article, weighted and termination constants, $\beta = 2\ \varepsilon = 10^{-17}$ (*Hu & Chan, 2015*), are selected when the performance of the fuzzy algorithm for theme features becomes relatively stable (*Naderipour, Fazel Zarandi & Bastani, 2022*). Figure 7 shows the density change of theme features when the number of feature points z ranges from 2 to 13. It is seen that when z is assigned to 6, the density of community theme features is 0.28 to achieve local optimization. When the number of community theme feature points reaches 6, the aggregation effect of users' theme vectors in the community becomes the best. Therefore, the number of feature points z takes the value of 6 in the subsequent experiment.

### Performance analysis of topic-based hierarchical community detection

To evaluate the performance of the ST-L algorithm, the following community detection models are used to compare.

GEMSEC (*Rozemberczki et al., 2019*): Community single-layer self-clustering based on machine learning to regularize users' social attributes.

vGraph (*Sun et al., 2019*): Based on the probability generation model that implements users' information to predict interaction probability, single-layer community detection.

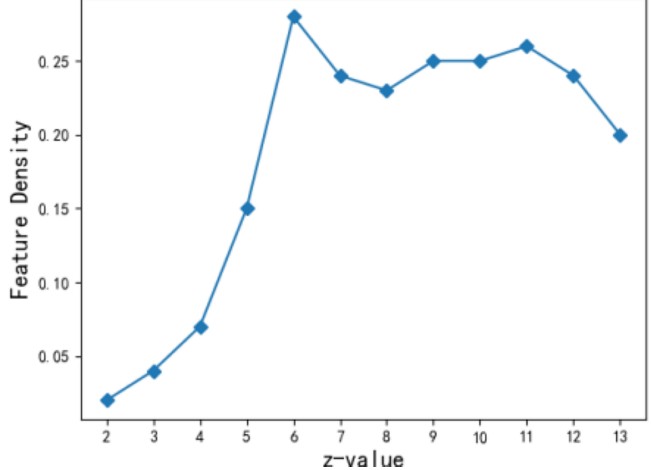

**Figure 7** Effect of *z* value on density of characteristic.

**Table 1** NMI index and modularity index of each algorithm.

|  | Weibo | | Zhihu | | Aminer | | Cora | |
|---|---|---|---|---|---|---|---|---|
|  | NMI | Modularity | NMI | Modularity | NMI | Modularity | NMI | Modularity |
| GEMSEC | / | 0.536 | / | 0.528 | 0.352 | 0.653 | 0.345 | 0.679 |
| vGraph | / | 0.579 | / | 0.570 | 0.284 | 0.710 | 0.366 | 0.735 |
| UICD | / | 0.498 | / | 0.503 | 0.298 | 0.689 | 0.315 | 0.702 |
| HIOC | / | 0.617 | / | 0.611 | 0.314 | 0.709 | 0.442 | 0.731 |
| ReinCom | / | 0.648 | / | 0.653 | 0.459 | 0.742 | 0.574 | 0.741 |
| ST-L | / | 0.650 | / | 0.659 | 0.527 | 0.756 | 0.589 | 0.757 |

UICD (*Jiang et al., 2022*): Based on UGC that obtains users' interest through the LDA algorithm and uses interest for single-layer community detection.

HIOC (*Zheng et al., 2019*): Based on corpora fixed label classification, hierarchical community detection is carried out by user interest and label similarity.

ReinCom (*Ding et al., 2021*): Users' characteristics are learned based on a deep learning model, optimizing community tree, and hierarchical community detection algorithm.

Since Weibo and Zhihu data sets have no preset classification, they cannot obtain the *NMI* index, so the *NMI* index is only discussed in the Aminer and Cora data sets. The detailed experimental data are shown in Table 1. The six algorithms have better effects on the Aminer and Cora data sets than on the Weibo and Zhihu data sets. Because Aminer and Cora are public data sets with preset classifications, users have high convergence according to classification. At the same time, for the four data sets, the *NMI* and modularity performance of the hierarchical community detection method is significantly superior to those of the single-layer community detection algorithm, indicating that the adoption of hierarchical community detection is effective. Among the hierarchical community detection algorithms, ST-L and ReinCom are superior to the HIOC algorithm, with fixed classification labels and multiple user detection in each data set. When

**Table 2 Performance of community alignment.**

|  | Accuracy |
|---|---|
| CAlign | 0.672 |
| PERFECT | 0.695 |
| MGA | 0.713 |

compared with ReinCom in Aminer and Cora, respectively, the ST-L has increased the *NMI* index by 14.8% and 2.6%, while the modularity index has increased by 1.8% and 2.1%. In Weibo and Zhihu, the modularity index improved slightly. In general, the ST-L makes full use of subject features and adopts the non-fixed label layering mode to improve the performance of the data set.

## The performance analysis of the community alignment

To better evaluate the community alignment performance of the MGA, this article compares it with the following algorithm on the microbot-Zhihu dataset.

CAlign (*Chen et al., 2017*): Based on user attributes, Dirichlet distributed clustering is used to build communities, and cross-network community alignment is performed.

PERFECT (*Sun et al., 2020*): The cross-network user information is embedded into the hyperbolic space based on the Poincare sphere model, clusters the users within the hyperbolic distance threshold into communities, and aligns the communities by the hyperbolic distance of each user.

Detailed experimental data are shown in Table 2. The MGA improves the accuracy of community alignment by 6.1% and 2.5%, respectively, when compared with CAlign and PERFECT algorithms, which proves that the MGA adopts multi-granularity fuzzy topic feature alignment and effectively reduces the influence of edge users on community alignment accuracy when compared to forced clustering method (*Naderipour, Fazel Zarandi & Bastani, 2022*).

## The performance analysis of user alignment

To better evaluate the MGA user alignment performance, this article will use the mainstream user alignment algorithm to compare the micro-blog Zhihu dataset.

BSNA (*Yuan et al., 2021*): Based on the BP neural network, the user name is uniformly mapped, the problem is transformed into a vector mapping problem, and the user alignment is performed.

PERFECT (*Sun et al., 2020*): The cross-network user information is embedded into hyperbolic space based on the Poincare ball model and aligns users by hyperbolic distance between vectors.

UGCLink (*Gao et al., 2021*): UGC is modelled based on a convolutional neural network and aligns users by the relationship between content and time.

MEgo2Vec (*Zhang et al., 2018*): Self-centred network based on graph neural network mining user attributes and structure for user alignment.

**Table 3 Performance of user alignment.**

| | Weibo-Zhihu | | |
| --- | --- | --- | --- |
| | Precision | Recall | F1 |
| BSNA | 0.572 | 0.495 | 0.530 |
| PERFECT | 0.646 | 0.534 | 0.581 |
| UGCLink | 0.652 | 0.541 | 0.587 |
| MEgo2Vec | 0.667 | 0.556 | 0.593 |
| MUIUI | 0.674 | 0.559 | 0.598 |
| MGA | 0.697 | 0.571 | 0.613 |

MUIUI (*Zeng et al., 2021*): Based on the three characteristics of user attributes, generated content, and relationships, a fusion classifier carries out user alignment.

The precision, recall, and F1 values of each comparison algorithm and the MGA algorithm are shown in Table 3. It was found that the performance of the BSNA algorithm for vector mapping alignment based only on user name is inferior to other algorithms, indicating that the accuracy of the alignment method based on a single feature is low. Similarly, the UGCLink algorithm based on UGC combined with time features and the PERFECT algorithm based on user topology embeddings that are combined with a small number of attribute features have improved performance, but the MEgo2Vec and MUIUI algorithms are significantly superior to the previous two algorithms. When compared with MEgo2Vec and MUIUI, respectively, the proposed MGA algorithm has improved the accuracy rate by 4.5% and 3.4%, the recall rate by 2.7% and 2.1%, and the F1 value by 3.4% and 2.5%, respectively. As the MGA fully utilizes the topic features generated by UGC, performing community detection before user alignment reduces the interference of user pairs with similar attribute features to a certain extent. The MGA has richer granularity choices than the two multi-granularity alignment algorithms.

The experimental results demonstrate that the MGA algorithm fully leverages UGC and integrates time characteristics comprehensively. It acquires the user stability theme, conducts hierarchical community detection, and performs cross-network multi-granularity alignment. The algorithm successfully addresses the issue of excessive users at the community edge, leading to improved accuracy, recall rate, and F1 values in user alignment by incorporating user attribute characteristics.

## CONCLUSION

This article, based on UGC, fully utilizes time characteristics to filter out short-term topics, acquire user-stable topics, enhance the Louvain algorithm, perform multi-level community detection for users without pre-set label classification, and introduces a multi-granularity alignment method for both group and user alignment modes. The improved fuzzy topic feature acquisition algorithm is adopted to resolve the problem of the reduced accuracy of forced clustering of edge users effectively (*Naderipour, Fazel Zarandi & Bastani, 2022*). In

the real data set, it is verified that the MGA has better performance in the multi-granularity alignment mode, covering both community and user alignments.

In future work, the edge weights based on stable topics can be integrated into the user structure features and attribute features to make the weights more diverse and further improve the performance of hierarchical community detection. The multi-granularity alignment method can also be improved by integrating structure, attributes, and content, reducing the impact of edge users on community alignment and further improving the accuracy of user alignment.

### Funding
This study was supported by the University of Shanghai for Science & Technology Natural Science Foundation Cultivation Project (20ZRPY08). The funders had no role in study design, data collection and analysis, decision to publish, or preparation of the manuscript.

### Grant Disclosures
The following grant information was disclosed by the authors:
University of Shanghai for Science & Technology Natural Science Foundation Cultivation Project: 20ZRPY08.

### Competing Interests
The authors declare that they have no competing interests.

### Author Contributions
- Jing Lu conceived and designed the experiments, performed the experiments, analyzed the data, performed the computation work, prepared figures and/or tables, authored or reviewed drafts of the article, and approved the final draft.
- Qikai Gai conceived and designed the experiments, performed the experiments, analyzed the data, performed the computation work, prepared figures and/or tables, authored or reviewed drafts of the article, and approved the final draft.

### Data Availability
The data and code are available in the Supplemental Files.

### Supplemental Information
Supplemental information for this article can be found online at http://dx.doi.org/10.7717/peerj-cs.1892#supplemental-information.

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
