# Peer review of "Multi-grained alignment method based on stable topics in cross-social networks"

_PeerJ Computer Science, doi:10.7717/peerj-cs.1892_

## Round 0.1 · original submission · Major Revisions

Dear authors
Thanks for your submission, unfortunately, the current version of the paper needs much improvement as suggested by the experts in the field, therefore you are required to carefully update the paper and resubmit.

Please also clearly explain what you mean by the cross social networks.

Highlight the contribution of your work

**Language Note:** The review process has identified that the English language must be improved. PeerJ can provide language editing services - please contact us at copyediting@peerj.com for pricing (be sure to provide your manuscript number and title). Alternatively, you should make your own arrangements to improve the language quality and provide details in your response letter. – PeerJ Staff

·

Basic reporting

The authors have proposed a multigrain alignment method for user and group identification across multiple social networks. Although the proposed method seems to have significance, the article organization, reporting style, and English language require major revisions even though the abstract is written very poorly, and the introduction and literature review is almost missing. With so many issues as above, it is very hard for me to confirm the validity of the methods and results. I feel that the article could be a good contribution only if the issues are resolved and properly proofread by a fluent English speaker so that reviewers can properly acknowledge the methods and findings.

The following are my observations on the article.
1. Abstract:
a) The abstract is quite weak and hardly represents the research. It should summarize the background of the topic, motivation of the research, current state-of-the-art, issue/challenges, and then the proposal to address them along with their significance.
b) Acronyms should be avoided from being used or defined in the abstract.

2. Reporting Style and English Language
I feel authors need to give considerable attention to the reporting style as there are several issues in terms of the correctness of the sentences, usage of articles, and punctuation, as well as the improper usage of acronyms such,

a) Lines 30 and 56 are very large sentences, and it seems they hardly convey any meaningful message. Authors should split the sentences into shorter ones.
b) Line 38, BP is used but never defined. Line 80, ST-L, is never defined and is not the correct way for acronyms. Line 140, TD-IDF
c) In-text citations are improper, and it is unclear what authors want to refer to it. For example, line 33 [2] cites what statement.
d) The sentence on line 46 uses anchor users two times.
e) Line 48, "probabilistic generating graph calculation," is unclear
f) The statement starting from line 51 and line 67 needs a proper reference.
g) In-text citations must be according to PeerJ Computer Science Journal specifications.
h) The article layout is not correct; authors should download the template from the submission page and use it to write the article.
i) The contributions starting from line 74 are not supported with a sufficient level of literature review /analysis/comparison. It is suggested that authors provide a comparison of the current state of the art related to the problem being resolved.
j) Avoid defining the equations in text and use equation numbering; furthermore, writing equations without defining what terms refer to is of no use to readers. This issue exists all over the article.

1. Comparison and analysis of the current state of the art is missing.

Experimental design

1 - The experiments are neither organized correctly not presented well. It is difficult for reviewers to validate the experimental setup.

3. The purpose and usage of fuzzy center clustering should be clearly described and discussed with suitable references.

4. Which data set was used for validation, and how were the experiments conducted? It is not clear how the data and code provided in supplementary files are related to the author's discussions in the article.

Validity of the findings

1 - Discussion part is missing and findings are not presented in correct way.

2 - The conclusion should provide quantitative improvements rather than just statements.

Additional comments

There are potential contributions in the paper, but its poor reporting style, numerous mistakes and absence of good organization of the contents make it very difficult to validate the experiments, results and findings.

Authors should explain how the supplementary files are related to article contents and add a discussion sections with quantitative comparison and analysis with other state of the art methods.

Reviewer 2 ·

Basic reporting

The research conducted has both linguistic and technical problems in its manuscript.

A thorough inspection should be done to make the content easier to read. We do not list every single one of them. A seasoned editing service will be very beneficial.

Experimental design

The article's technical material is intriguing and adds to the body of literature. Nonetheless, a few problems negatively impact the article's quality. We thus go over them in depth. Authors are expected to address them in detail to clear up any ambiguity that may arise in certain areas.

1. The article's algorithms and suggested techniques should be provided in an understandable and comprehensible way.
2. Every definition needs to be verified and stated precisely.
3. Every mathematical formula needs to be referenced inside the text. There should be clarity in the vocabulary used for those. A few of them are deceptive. The writers ought to review and correct them.
4. The article's contribution needs to be emphasized further

Validity of the findings

1. Was the authors' fuzzy center clustering approach used? If not, kindly reference it.
2. How did the writers determine whether the information fit the fuzzy-based approach? Talk about it, please.
3. c01: {1,3,9,10,15}, c02: {2, 6, 12}, c03: {4,5,11,13,17}, c04: {7,8,14,16}, – how did the writers obtain these? Please talk about this further.
4. How did the writers determine the fuzzy cluster numbers? Please talk about this further.
5. What leads the authors to believe that the data is compatible with the fuzzy-based method? Please talk about this further.
6. How were sample data used by the authors? Which technique of sampling was used? Please talk about this further.
7. How does the author determine k when they perform k-means clustering? Please talk about this further.
8. What prevented the authors from comparing the outcomes using a benchmark dataset? If the author can do it to make the article better, we advise them to do so.
9. The benefits and drawbacks of the suggested approach should be discussed in the conclusion section.

Cite this review as

---

## Round 0.2 · accepted · Accept

Based on the input from the experts, the revised version of the paper is acceptable by the reviewers, therefore I'm pleased to inform you that your manuscript has been recommended for publication

·

Basic reporting

The authors have improved the reporting style of article significantly.

Experimental design

Experimental design is good and authors are covered all aspects of evaluation with real dataset.

Validity of the findings

Valid

Additional comments

Although the article is now ready of publication based on the validity of methods, experiments and findings.

Reviewer 2 ·

Basic reporting

I am pleased to report that the authors have diligently and satisfactorily addressed all the suggestions for changes, modifications, and improvements that were made during the previous review process. Their commitment to refining the manuscript has significantly strengthened the overall quality and clarity of the content.

I believe the revised version now aligns well with the standards and guidelines of PeerJ Computer Science. Please consider this communication as confirmation that the authors have successfully incorporated the recommended revisions. I believe the manuscript is now ready for the next stage of the review process.

Experimental design

I am pleased to report that the authors have diligently and satisfactorily addressed all the suggestions for changes, modifications, and improvements that were made during the previous review process. Their commitment to refining the manuscript has significantly strengthened the overall quality and clarity of the content.

I believe the revised version now aligns well with the standards and guidelines of PeerJ Computer Science. Please consider this communication as confirmation that the authors have successfully incorporated the recommended revisions. I believe the manuscript is now ready for the next stage of the review process.

Validity of the findings

I am pleased to report that the authors have diligently and satisfactorily addressed all the suggestions for changes, modifications, and improvements that were made during the previous review process. Their commitment to refining the manuscript has significantly strengthened the overall quality and clarity of the content.

I believe the revised version now aligns well with the standards and guidelines of PeerJ Computer Science. Please consider this communication as confirmation that the authors have successfully incorporated the recommended revisions. I believe the manuscript is now ready for the next stage of the review process.

Cite this review as